

# An ambiguous N-terminus drives the dual targeting of an antioxidant protein Thioredoxin peroxidase (TgTPx1/2) to endosymbiotic organelles in *Toxoplasma gondii*

Pragati Mastud and Swati Patankar

Department of Biosciences and Bioengineering, Indian Institute of Technology Bombay, Powai, Mumbai, India

## ABSTRACT

*Toxoplasma gondii* harbors two endosymbiotic organelles: a relict plastid, the apicoplast, and a mitochondrion. The parasite expresses an antioxidant protein, thioredoxin peroxidase 1/2 (TgTPx1/2), that is dually targeted to these organelles. Nuclear-encoded proteins such as TgTPx1/2 are trafficked to the apicoplast via a secretory route through the endoplasmic reticulum (ER) and to the mitochondrion via a non-secretory pathway comprising of translocon uptake. Given the two distinct trafficking pathways for localization to the two organelles, the signals in TgTPx1/2 for this dual targeting are open areas of investigation. Here we show that the signals for apicoplast and mitochondrial trafficking lie in the N-terminal 50 amino acids of the protein and are overlapping. Interestingly, mutational analysis of the overlapping stretch shows that despite this overlap, the signals for individual organellar uptake can be easily separated. Further, deletions in the N-terminus also reveal a 10 amino acid stretch that is responsible for targeting the protein from punctate structures surrounding the apicoplast into the organelle itself. Collectively, results presented in this report suggest that an ambiguous signal sequence for organellar uptake combined with a hierarchy of recognition by the protein trafficking machinery drives the dual targeting of TgTPx1/2.

## INTRODUCTION

Endosymbiotic organelles, plastids and mitochondria, carry out specialized and essential functions in eukaryotes. The proteins required for these functions are predominantly nuclear-encoded and are trafficked from their site of synthesis in the cytosol to the organelles by different pathways. To avoid mis-localization, these nuclear-encoded organellar proteins have specific targeting signals (*Kunze & Berger, 2015*) that are recognized by organellar translocons located on the membranes (*Schleiff & Becker, 2011*). While many proteins localize to single organelles due to the presence of these specific signals, there are increasing reports of dual targeting (*Karniely & Pines, 2005*; *Hirakawa, Burki & Keeling, 2012*;

Corresponding author
Swati Patankar, patankar@iitb.ac.in

*Carrie & Small, 2013*; *Baudisch et al., 2014*), mostly from plants, where certain proteins are localized to both the chloroplasts and mitochondria (*Chew et al., 2003*; *Carrie & Small, 2013*; *Baudisch et al., 2014*). Dual targeting of proteins can be achieved through alternative splicing, or the utilization of alternative transcription initiation sites, where the two gene products reach different locations (*Danpure, 1995*; *Small et al., 1998*). Additionally, an interesting mechanism observed for dual targeting to the endosymbiotic organelles is that of a single translational product localizing to both chloroplasts and mitochondria (*Chew et al., 2003*; *Karniely & Pines, 2005*; *Pino et al., 2007*; *Berglund et al., 2009*; *Baudisch & Klösgen, 2012*). In such cases, it has been proposed that ambiguous targeting signals have dual specificities that result in recognition by translocons on both organelles (*Chew et al., 2003*).

Dual targeting of proteins to endosymbiotic organelles has also been observed in the phylum Apicomplexa, which includes the obligate endo-parasitic pathogens of the *Plasmodium spp.* and *Toxoplasma gondii*. Apicomplexan parasites possess two organelles of endosymbiotic origin: the mitochondrion, and a relict plastid, the apicoplast. Unlike chloroplasts, apicoplasts have four membranes (*Lemgruber et al., 2013*), and are thought to have been acquired by the secondary endosymbiosis of eukaryotic red algae (*Wilson, 1993*; *McFadden et al., 1996*; *Blanchard & Hicks, 1999*; *Waller et al., 2003*; *Janouskovec et al., 2010*). Both the apicoplast and mitochondrion are hubs for essential metabolic pathways (*Vercesi et al., 1998*; *Wilson, 2002*; *Seeber, 2002*; *Gardner et al., 2002*; *Uyemura et al., 2004*; *Sato et al., 2004*; *Van Dooren, Stimmler & McFadden, 2006*; *Sheiner, Vaidya & McFadden, 2013*). To carry out these indispensable functions, nuclear-encoded proteins are trafficked from their site of synthesis in the cytosol to these organelles (*Cooke et al., 2004*; *Sheiner & Soldati-Favre, 2008*).

Proteins destined for the apicoplast and mitochondria contain N-terminal targeting sequences that are cleaved on protein import. Proteins targeted to the apicoplast possess an N-terminal bipartite sequence (BTS) (*Waller et al., 1998*; *Waller et al., 2000*; *DeRocher et al., 2000*; *Yung, Unnasch & Lang-Unnasch, 2001*; *Zuegge et al., 2001*; *Sheiner et al., 2015*). This BTS comprises of an N-terminal signal peptide followed by a plant-like transit peptide (*Waller et al., 2000*; *DeRocher et al., 2000*). The signal peptide directs the nuclear-encoded apicoplast targeted proteins to the endoplasmic reticulum co-translationally, where it gets cleaved to expose the transit peptide. The transit peptide then assists in trafficking of the apicoplast-targeted protein from the endoplasmic reticulum to the apicoplast. On the other hand, for mitochondrial transport in *T. gondii* and *P. falciparum*, nuclear-encoded proteins possess a single mitochondrial targeting sequence (MTS), generally present at their N-terminus (*Toursel et al., 2000*; *Garrison & Arrizabalaga, 2009*; *Sheiner et al., 2015*; *Mallo et al., 2018*). The MTS governs the import of proteins to the mitochondrion via uptake by translocons (*Van Dooren, Stimmler & McFadden, 2006*; *Van Dooren et al., 2016*). Unlike apicoplast trafficking, mitochondrial proteins are imported post-translationally without the involvement of an endoplasmic reticulum.

Despite two different routes being employed for trafficking i.e., the secretory pathway for apicoplast-targeted proteins and translocons mediated post-translational uptake for mitochondrial targeting proteins, there are a few reports documenting the dual

targeting of proteins to both these organelles in *T. gondii* and *P. falciparum* (*Pino et al., 2007*; *Günther et al., 2007*; *Saito et al., 2008*; *Read et al., 2010*; *Chaudhari, Narayan & Patankar, 2012*). Prediction of organellar targeting signals in the proteins targeted dually to the apicoplast and the mitochondrion in apicomplexan parasites reveal that *P. falciparum* serine hydroxyl methyl transferase (PfSHMTm) possesses a MTS (*Read et al., 2010*), *P. falciparum* glutathione peroxidase-like thioredoxin peroxidase (PfTPx$_{Gl}$) and *T. gondii* pyruvate kinase (TgPyKII) display signal anchor sequences (*Saito et al., 2008*; *Chaudhari, Narayan & Patankar, 2012*; *Narayan et al., 2018*), *T. gondii* superoxide dismutase (TgSOD2), thioredoxin peroxidase (TgTPx1/2) and aconitase (TgACN) harbor BTS and MTS (*Pino et al., 2007*) while *P. falciparum* serine hydroxyl methyl transferase (PfSHMTc) and lipoate protein ligase 2 (PfLplA2) lack any of the detectable organellar targeting signals (*Günther et al., 2007*; *Read et al., 2010*). This suggests that the signals for dual targeting to apicoplast and mitochondrion vary from each other and show no specific pattern in common.

Of the dually targeted proteins studied, a single translation product of TgSOD2 and TgACN is targeted to the apicoplast and the mitochondrion with the help of N-terminal apicoplast BTS and MTS. In contrast, TgPyKII utilizes an alternative translation initiation site to generate two proteins with different N-terminii. A further study on TgSOD2 revealed that a single point mutation in the N-terminus of a TgSOD2 GFP fusion protein shifts its localization from the mitochondrion to the apicoplast (*Brydges, 2003*). Clearly, open questions remain; does the dual targeting of proteins to the apicoplast and mitochondrion occur due to superimposed signals and if so, which signals?

To understand mechanisms underlying dual targeting, we studied the antioxidant protein, thioredoxin peroxidase 1/2 (TgTPx1/2), an alternatively spliced form of *T. gondii* thioredoxin peroxidase, TgTPx. TgTPx1/2 localizes to both the apicoplast and mitochondrion in *T. gondii* parasites (*Pino et al., 2007*), reflecting the generation of oxidative radicals in these sub-cellular compartments and their need to handle oxidative stress. Through a combination of bioinformatics and mutational analyses we show that an N-terminal ambiguous sequence directs the dual targeting of TgTPx1/2 to the apicoplast and mitochondrion. This sequence shows an overlap between the signal peptide of the apicoplast targeting signal and the predicted amphipathic helix of the mitochondrial targeting signal. Interestingly, proteins with mutants in the N-terminus of TgTPx1/2 are found in either organelle while a truncated N-terminus results in localization of a reporter protein to punctate structures close to, but not overlapping with the apicoplast. These data led us to propose that the organellar receptors compete for the ambiguous N-terminus of TgTPx1/2 and this N-terminus consists of an ambiguous sequence where the signals for localization to each organelle are overlapping yet separable.

## MATERIALS AND METHODS

### *T. gondii* culture, transfection and selection

*T. gondii* parasites (wild type RH strain) were grown in primary Human Foreskin fibroblasts (HFF-1) cells. HFF-1 cells were obtained from ATCC and grown in Dulbecco's Modified

Eagle's Medium (Gibco[TM], Mfr. No. 12100-046) containing 3.7 g/L sodium bicarbonate and 2.38 g/L HEPES, supplemented with 10% Cosmic calf serum (HyClone[TM]) and 20 mg/L Gentamicin. Transient transfections of extracellular *T. gondii* parasites were carried out at 1.5 kV, 50 Ω and 25 μF using Bio-Rad GenePulser Xcell system and subjected to immunostaining 24 h post transfection. Stable lines were generated by Restriction Enzyme Mediated Integration (REMI) using NotI followed by selection of the parasites for growth in 20 μM chloramphenicol.

## Plasmid construction

Plasmid pCTG-EGFP was modified to express 1X HA tag (modified plasmid pCTG-HA) by excising the GFP and inserting the annealed oligos pCTG-HA(F) and pCTG-HA(R), containing the coding sequence for HA between the NheI and PstI sites (NheI site modified in pCTG-HA). Total RNA was isolated from *T. gondii* RH strain using TRIsoln (Merck Genei). The cDNA for TgTPx1/2 (ToxoDB gene ID: TGME49_266120) was generated using gene specific reverse primer TPx1/2R by RevertAid Reverse Transcriptase (Thermo Scientific). TgTPx1/2 was amplified from the cDNA using primers TPx1/2F, TPx1/2R and cloned between MfeI and NdeI sites in the vector pCTG-HA to generate TgTPx1/2-HA. TgTPx1/2(M10A) was constructed using TPx1/2M10AF and TPx1/2R and cloned between MfeI and NdeI sites in the vector pCTG-HA. To generate amino acid mutants TgTPx1/2(R24A), TgTPx1/2(L17A,L27A) and TgTPx1/2(E15A,R24A), site-directed mutagenesis was carried out to replace the respective amino acids with alanine. To generate various N-terminal fusions of TgTPx1/2 with EGFP, regions of the coding sequence from 1–150, 1–120 and 1–90 bp were amplified using TPx1/2GFPF as a common forward primer and TPx1/2N1-50-GFPR, TPx1/2N1-40-GFPR and TPx1/2N1-30-GFPR as the respective reverse primers. The amplified region was cloned between the NheI and AvrII sites of the plasmid pCTG-EGFP. To generate signal peptide deletion mutants TgTPx1/2(Δ2–28) and TgTPx1/2(Δ2–32), site-directed mutagenesis were carried out to delete the amino acids stretch from 2–28 and 2–32 respectively. Sequences of oligonucleotides and primers used for cloning are given in the Table 1. All the resultant plasmids expressed the fusion proteins under the control of the α-tubulin promoter and were sequenced to confirm the inserts and the mutations. The plasmid pCTG-EGFP was obtained as a kind gift from Prof. Shobhona Sharma, Tata Institute of Fundamental Research, Mumbai, and the plasmids expressing SPTP-SOD2-GFP, SPTP-SOD2-DsRed were obtained as a kind gift from Prof. Dominique Soldati-Favre, University of Geneva, Geneva.

## Microscopy

HFF-1 cells grown in chamber slides (SPL Life Sciences Co.) were infected with *T. gondii* parasites for microscopy. Transient transfectants were processed for immunostaining 24 hours post-transfection. The parasites were fixed with 4% formaldehyde and 0.0075% glutaraldehyde, followed by permeabilization with 0.25% Triton X-100 for 10 min and then blocked with 3% BSA-PBS for an hour. The parasites were incubated for 2 h at room temperature with the primary antibodies 1:1000 anti-HA (C29F4) Rabbit monoclonal antibody (mAb) from Cell Signaling Technology[®], 1:1000 anti-HA (monoclonal, Clone

**Table 1  Sequences of oligonucleotides and primers used for cloning are given in the Table 1.** Restriction sites are underlined and the mismatch regions are in bold letters. All primers are in the 5′ to 3′ direction.

| Construct | Forward primer | Reverse primer |
|---|---|---|
| pCTG-HA | pCTG-HA(F) CTAGTCAATTGCGCCGGCGCCCGGGCATATG GGCGGCTACCCTTACGACGTCCCTGACTA CGCGTAACTGCA | pCTG-HA(R) GTTACGCGTAGTCAGGGACGTCGTAAGGGTAG CCGCCCATATGCCCGGGCGCCGGCGCAATTGA |
| TgTPx1/2-HA | TPx1/2F TACACAATTGGACAAAATGCTTCCTCTCTGCGCGTCT | TPx1/2R GGATCATATGGAGAGGCTTCAGCTCGCCTG |
| TgTPx1/2N$_{1-50}$-EGFP | TPx1/2GFPF TACAGCTAGCGACAAAATGCTTCCTCTCTGCG | TPx1/2N1-50-GFPR TAATCCTAGGGCCGCCGCCGCCGTGACC AGCAGTCC |
| TgTPx1/2N$_{1-40}$-EGFP | TPx1/2GFPF TACAGCTAGCGACAAAATGCTTCCTCTCTGCG | TPx1/2N1-40-GFPR TAATCCTAGGGCCGCCGCCCCAAAAAGCG CGACCGTAC |
| TgTPx1/2N$_{1-30}$-EGFP | TPx1/2GFPF TACAGCTAGCGACAAAATGCTTCCTCTCTGCG | TPx1/2N1-30-GFPR TAATCCTAGGGCCGCCGCCACATGTAGCCA AGCAAAG |
| TgTPx1/2(Δ2-28) | TgTPx1/2(Δ2-28)F ACATGTTTGGGGGCGTTGTACGGTCGCG | TgTPx1/2(ΔSP)R CATTTTGTCCAATTGACTAGCAGATCTAAAAGGG |
| TgTPx1/2(Δ2-32)F | TgTPx1/2(Δ2-32)F GCGTTGTACGGTCGCGCTTTTTGGCGG | TgTPx1/2(ΔSP)R CATTTTGTCCAATTGACTAGCAGATCTAAAAGGG |
| TgTPx1/2(M10A) | TPx1/2M10AF TACACAATTGGACAAAATGCTTCCTCTCTGCGCG TCTTCTCAGGCGTATCTCGG | TPx1/2R GGATCATATGGAGAGGCTTCAGCTCGCCTG |
| TgTPx1/2(R24A) | TPx1/2R24AF CAGTGCTGCCG**GCG**CTTTGCTTGGCTACATG | TPx1/2R24AR GGCAAGACAGGCACTCCTGTCCGAGATACATC |
| TgTPx1/2(L17A,L27A) | TPx1/2L17AF CTGCCGCGTCTTTGC**GCT**GCTACATGTTTG | TPx1/2L27AR CACTGGGCAAGA**CGC**GCACTCCTGTC |
| TgTPx1/2(E15A,R24A) | TPx1/2E15AR24AF CAGTGCTGCCG**GCG**CTTTGCTTGGCTAC | TPx1/2E15AR24AR GGCAAGACAGGCA**CGC**CTGTCCGAGATACATC |

HA-7, mouse IgG1 Isotype, from Sigma Aldrich®) or 1:1000 anti-ACP antibodies (obtained as a kind gift from Prof. Dhanasekaran Shanmugam, NCL, Pune) followed by three washes of 1X PBS. The parasites were subjected to incubation with secondary antibody (1:400 anti-rabbit Alexa 568, 1:400 anti-rabbit Alexa 488, 1:250 anti-mouse Alexa 594 from Invitrogen) for 1.5 h followed by three washes with 1X PBS for 10 minutes each. The samples were mounted in ProLong Diamond Antifade mountant and stored at 4 °C. Intracellular parasite mitochondrial staining was carried out using the protocol previously described (*Salunke et al., 2018*) with the fluorescent dye MitoTracker® Red CMXRos (300 nM) from Invitrogen™.

The images were captured by Zeiss LSM 780 NLO confocal microscope using a Plan-Apochromat 100X, 1.4 NA objective. Optical sections were captured at the interval of 0.44 μm and each planar image was averaged two times for image acquisition. Maximum intensity projection images for these optical sections were obtained. For clarity, the images were processed later for brightness and contrast using ImageJ. No non-linear adjustments such as changes to the gamma settings and curves were performed. For each transfectant,

approximately 50 parasite rosettes were observed microscopically to ensure consistency in the morphology observed and approximately 10 rosettes were captured for documentation. Representative images for each transfectant have been shown in the figures.

To measure Pearson's correlation coefficient (PCC) for different mutants with the organellar markers, 'Coloc' tool of the Imaris software was employed. Unprocessed Z-stack images were utilized for measuring the PCC. A region of the organellar marker was cropped and tested for co-localization of protein of interest. After cropping the desired region of the organelle from the entire Z-stack, an 'Automatic thresholding' option was used to set the threshold for both the channels from the control panel. After building the 'Coloc channel', the 'Coloc volume statistics' were observed and the Pearson's coefficient in co-localized volume were used to interpret co-localization. A dot plot (Fig. S1) displaying the PCC generated for the parasites lines expressing the individual constructs with the appropriate markers was plotted using the GraphPad Prism version 8.0.1 (244) for Windows, GraphPad Software, La Jolla California USA. The average PCC values derived from this graph have been mentioned next to each panel in the figures. In this study, an average PCC value from 0.3–1.0 has been termed as co-localization, a PCC value between −1 to 0.15 has been termed as a lack of co-localization. Average PCC values falling between 0.15–0.3 have been termed as partial co-localization.

## Bioinformatics analysis

SignalP 3.0-HMM (*Nielsen & Krogh, 1998*; *Bendtsen et al., 2004*; *Cilingir, Broschat & Lau, 2012*) and MitoProt II (*Claros & Vincens, 1996*) were employed for predicting the signal peptide and MTS.

To identify the residues important for dual targeting of the protein, an *in silico* alanine scanning mutagenesis of the residues was carried out. Alanine is the choice of replacement here because it is an uncharged, chemically inert, non-bulky amino acid that does not impose any extreme electrostatic or steric effects. Secondly, alanine displays a deletion in the side chain at the $\beta$-carbon and yet does not alter the main chain conformation of the protein unlike glycine and proline.

For *in silico* alanine scanning mutagenesis study, full-length TgTPx1/2 protein was used as an input sequence in the two bioinformatics software. For each of the first 50 amino acids, the native non-alanine amino acid was replaced with alanine and the resulting full length proteins were used as input sequence. Default settings were employed for the analysis. Since the changes in SignalP and MitoProt values were observed for the amino acids in the region 1–30, we have plotted the graph displaying these values from residues 1–30.

α-helix predictions for proteins were carried out using the online computational tool Webgenetics (https://www.webgenetics.com/acts/wg?group=protein&prog=wheel) while the grand average of hydrophobicity (GRAVY) values were determined with the ExPASy-ProtParam tool (*Wilkins et al., 1999*).

## Western blotting

Post-transfection, parasites stably expressing the proteins under chloramphenicol drug pressure were lysed in SDS-PAGE gel loading buffer (1X) and an equal number of parasites

were loaded on a 15% SDS-PAGE gel. The proteins were transferred to PVDF membrane, blocked for an hour with 5% BSA-Tris Buffered Saline (TBS) and probed with anti-HA (C29F4) Rabbit monoclonal antibody (mAb) (Cell Signaling Technology®). Following this, the blot was incubated with HRP conjugated secondary antibodies (Merck) for 1 h and developed in 10 mM Tris-Cl (pH 7.6) with diaminobenzidine (DAB).

## RESULTS

### TgTPx1/2 is dually targeted to the apicoplast and the mitochondrion in *T. gondii* parasites

Previous reports have indicated that for tagging of a dually targeted protein TgSOD2, GFP fusions show altered organellar localization while small epitope tags do not (*Brydges, 2003*; *Pino et al., 2007*). Therefore, for studying the dually targeted protein TgTPx1/2, full-length TgTPx1/2 was fused with an HA epitope tag at the C-terminus. TgTPx1/2-HA localized to both the apicoplast and the mitochondrion in *T. gondii* parasites, as confirmed by co-localization with previously known mitochondrial and apicoplast markers SPTP-SOD2-GFP and acyl carrier protein (ACP) respectively (Figs. 1A, 1B). This result replicates data from a previous report (*Pino et al., 2007*) and confirms that TgTPx1/2 shows dual targeting to the endosymbiotic organelles in *T. gondii*.

Dual targeting can be achieved at the level of transcription where two mRNAs are formed from a single gene by differential splicing. Indeed, TgTPx1/1 and TgTPx1/2 are alternatively spliced forms of the TgTPx gene (*Pino et al., 2007*). In order to rule out the possibility of differential splicing being responsible for dual targeting, in this study, cDNA of TgTPx1/2 was amplified using primers that are specific to the TgTPx1/2 spliced form of the mRNA.

Proteins that are trafficked to endosymbiotic organelles are generally cleaved on import (*Waller et al., 1998*; *Toursel et al., 2000*; *Mallo et al., 2018*). To investigate the processing of TgTPx1/2, Western blot analysis of lysates from parasites expressing TgTPx1/2-HA was carried out using anti-HA antibodies. The blot revealed two bands of sizes ~40 kDa and ~26 kDa (Fig. 1C). The ~40 kDa band corresponds to a size equivalent or close to the full-length TgTPx1/2 with a single C-terminal HA tag. This band indicates that a fraction of TgTPx1/2 protein may not be processed. The presence of the ~26 kDa band, ~14 kDa less than the predicted full-length TgTPx1/2, suggests that the N-terminus might harbor signals that are cleaved in the apicoplast, mitochondrion or both organelles. Along with the two intense bands, faint bands between the 40 kDa and 26 kDa bands of TgTPx1/2 were also observed. These faint bands appear to be pre-processed versions of TgTPx1/2, cleaved for uptake in the apicoplast and/or the mitochondrion. This has been observed in case of other apicoplast and mitochondrial targeted proteins as well (*Pino et al., 2007*; *Seidi et al., 2018*).

### The N-terminus of TgTPx1/2 directs dual trafficking to the apicoplast and the mitochondrion

Previous studies have shown that signals for apicoplast and mitochondrial targeting are generally present at the N-terminus of proteins. Bioinformatics analyses suggest that

<p>PeerJ</p>

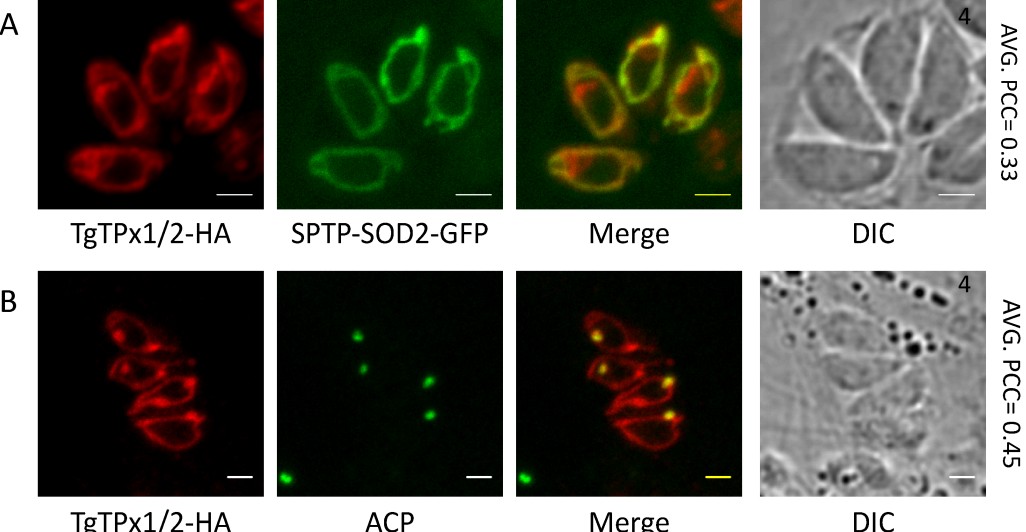

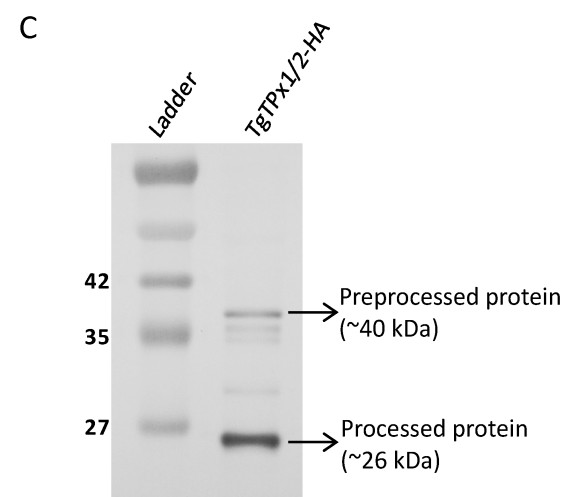

**Figure 1  Assessing the localization of a *T. gondii* antioxidant protein TgTPx1/2.** (A, B) Microscopic images of *T. gondii* parasites showing localization of an HA tagged TgTPx1/2 (red) with a mitochondrial marker SPTP-SOD2-GFP (green) and an apicoplast marker ACP (green). (C) A Western blot analysis for total cell lysate of parasites stably expressing TgTPx1/2-HA using anti-HA antibodies. Full-length Western blot is represented in Fig. S2. Scale bar, two μm. Numbers (in black) at the top right corner of the DIC+Merge panel indicates the number of parasites inside the vacuole.

TgTPx1/2 has a potential N-terminal signal peptide (SignalP 3.0-HMM score 0.527), a characteristic feature for the first step of apicoplast targeting. SignalP analysis further reveals a putative signal peptide cleavage site between residues 28–29 as per SignalP 3.0 NN and between residues 32–33 as per SignalP 3.0 HMM respectively. Along with the presence of a potential signal peptide, MitoProt analysis predicts a MTS with a potential cleavage

site between residues 100–101. This cleavage site prediction matches with the approximate molecular weight observed for the cleaved TgTPx1/2 from the Western blot analysis.

As the SignalP 3.0 prediction indicated a signal peptide with a cleavage site in the vicinity of 30 amino acids, the first 30 amino acids of TgTPx1/2 were fused with EGFP. TgTPx1/2N$_{1-30}$-EGFP displays GFP fluorescence around the parasite, suggesting a characteristic plasma membrane staining observed for proteins targeted to the plasma membrane (*Sheiner & Soldati-Favre, 2008*) (Figs. 2A, 2B). Proteins trafficking to the plasma membrane of the *T. gondii* parasite are known to be driven by the N-terminal signal peptide through a constitutive vesicular pathway mediated via the ER-Golgi network (*Karsten et al., 1998*; *Sheiner & Soldati-Favre, 2008*). Localization of TgTPx1/2N$_{1-30}$-EGFP to the plasma membrane indicates that the first 30 amino acids of TgTPx1/2 contain a signal peptide that can direct EGFP through the secretory route. Furthermore, TgTPx1/2N$_{1-30}$-EGFP neither shows any overlap with the apicoplast marker ACP nor with a mitochondrial marker MitoTracker Red (Figs. 2A, 2B).

To further confirm the presence of a signal peptide in amino acids 1-30 of TgTPx1/2, residues from 2–28 and 2–32 were deleted. These deletions were selected on the basis of the signal peptide cleavage predictions for TgTPx1/2 by SignalP 3.0 NN and SignalP 3.0 HMM respectively. Deletion of a predicted signal peptide [TgTPx1/2($\Delta$2–28)] from the BTS of apicoplast targeted proteins should abolish its targeting to the apicoplast completely, localizing the protein to the cytosol. Indeed, TgTPx1/2($\Delta$2–28) localizes to the cytosol of *T. gondii* parasites and does not co-localize with the apicoplast marker ACP (Fig. 2C) or the mitochondrial marker MitoTracker Red (Fig. 2D). Apart from the dense cytosolic staining, an occasional mitochondrion signal was observed in parasites transiently transfected with TgTPx1/2($\Delta$2–28).

Furthermore, TgTPx1/2($\Delta$2–32) shows no co-localization with an apicoplast marker ACP (Fig. 2E). The protein is found in the cytosol and shows a partial co-localization with the mitochondrial marker MitoTracker Red (Fig. 2F). The mitochondrial signal suggests that the amino acid residues from 32 onwards contain a MTS. The presence of a distinct cytosolic signal along with an absence of apicoplast localization demonstrates that the N-terminal region from 1–32 amino acids behaves like a signal peptide of the BTS that is essential for apicoplast targeting of TgTPx1/2.

For BTS-mediated trafficking of the apicoplast proteins, the signal peptide is followed by a transit peptide. In order to map the transit peptide of TgTPx1/2, downstream sequences were added incrementally to the signal peptide of the protein. When the 10 amino acids stretch downstream of the signal peptide (1-30 amino acids) was added to the signal peptide, the resultant fusion protein TgTPx1/2N$_{1-40}$-EGFP was shown to be accumulated in punctate structures surrounding the apicoplast (Fig. 3A). This suggests that the first 40 amino acids of TgTPx1/2 appear to contain a signal peptide and partial sequences of the apicoplast transit peptide.

Further addition of 10 residues to the 1-40 amino acid stretch resulted in localization of the fusion protein TgTPx1/2N$_{1-50}$-EGFP to both the apicoplast and mitochondrion, as confirmed by co-localization with a mitochondrial marker SPTP-SOD2-DsRed and a partial co-localization with an apicoplast marker, ACP (Figs. 3B, 3C). Along with the

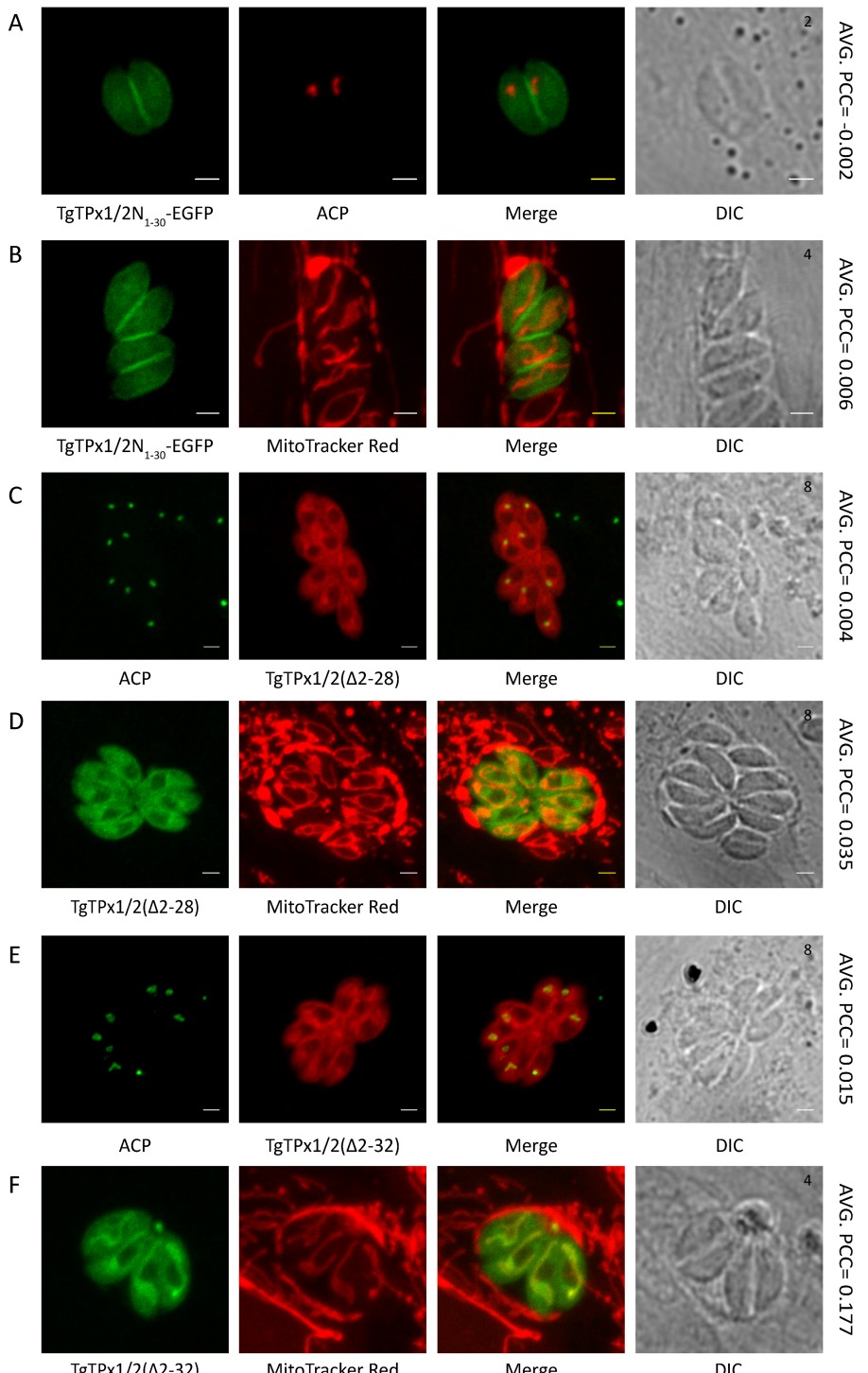

**Figure 2 Mutational analysis of the TgTPx1/2 N-terminus to understand the presence of the targeting signals.** Microscopic images of TgTPx1/2N$_{1-30}$-EGFP (green) where (A) apicoplast marker protein ACP (red) is used to label the apicoplast while (B) MitoTracker Red (red) is (continued on next page...)

**Figure 2 (…continued)**
used to label the mitochondrion. Immunofluorescence images of parasites expressing the signal peptide deletion mutants (C, D) TgTPx1/2($\Delta$2-28) (red /green) and (E, F) TgTPx1/2($\Delta$2-32) (red/green). Here ACP (green) and MitoTracker Red (red) are used to label the apicoplast and mitochondrion respectively. Scale bar, two $\mu$m. Numbers (in black) at the top right corner of the DIC panel indicates the number of parasites inside the vacuole.

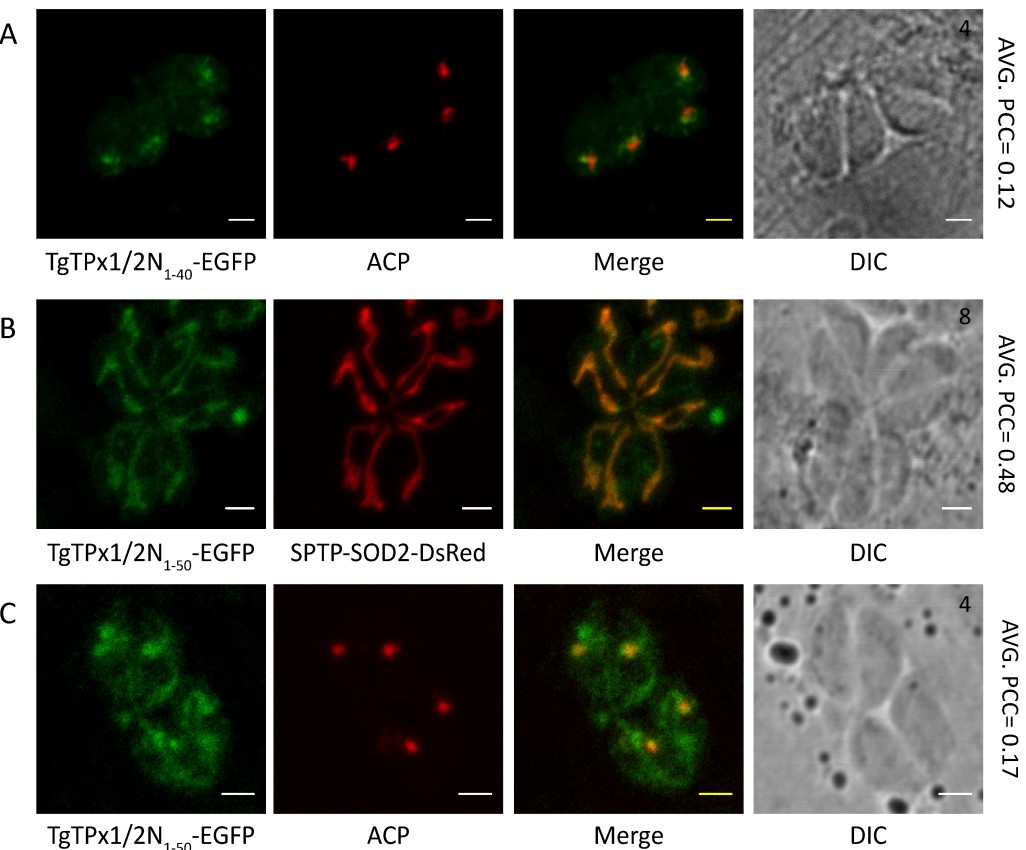

**Figure 3 Delineating the organellar targeting signals responsible for dual targeting of TgTPx1/2 to the apicoplast and mitochondrion.** Fluorescence images of parasites transiently expressing (A) TgTPx1/2N$_{1-40}$-EGFP (green) and (B, C) stably expressing TgTPx1/2N$_{1-50}$-EGFP (green). Here, ACP (red) is used as an apicoplast marker while with SPTP-SOD2-DsRed (red) is employed to label the parasite mitochondrion. Scale bar, two $\mu$m. Numbers (in black) at the top right corner of the DIC+Merge panel indicates the number of parasites inside the vacuole.

apicoplast and mitochondrial localization some amount of the fusion protein displaying a staining characteristic of the *T. gondii* plasma membrane and cytosol was also observed.

Collectively, the data on TgTPx1/2 N-terminal GFP fusions and signal peptide deletion mutants suggest that a signal peptide is present from amino acids 1-30 while the region from residues 30–50 encompasses signals for both apicoplast (transit peptide of the BTS) and mitochondrial trafficking (MTS).

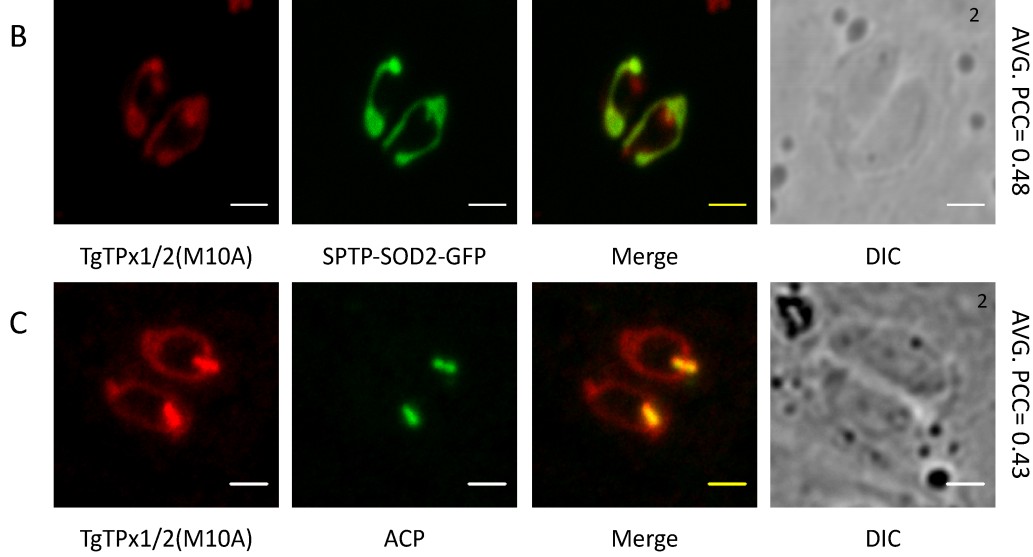

A       N-Terminus of TgTPx1/2

**M**LPLCASSQ**M**YLGQECLSCPVLPRLCLATCLGALYGRAFWRTNK-

**Figure 4 Analyzing the involvement of an alternative translational initiation site in dual targeting of TgTPx1/2 by mutational analysis.** (A) TgTPx1/2 has a second methionine at position 10 in the N-terminal 50 amino acids. Underlined sequence indicates the signal peptide predicted by SignalP 3.0-HMM. (B, C) Immunofluorescence images of parasites expressing TgTPx1/2(M10A) (red) with a mitochondrial marker SPTP-SOD2-GFP (green) and an apicoplast marker ACP (green) Scale bar, two μm. Numbers (in black) at the top right corner of the DIC panel indicates the number of parasites inside the vacuole.

## A single translational product of TgTPx1/2 directs the dual targeting of TgTPx1/2 to the apicoplast and the mitochondrion

Proteins that are dually targeted could have multiple translation initiation sites that result in isoforms with different N-terminal sequences (*Danpure, 1995*; *Small et al., 1998*; *Saito et al., 2008*). The first 50 amino acids of TgTPx1/2, that are demonstrated to harbour enough signals for dual targeting of TgTPx1/2 to the apicoplast and mitochondrion, contain one methionine at position 10 in addition to the methionine at position 1 (Fig. 4A), suggesting the possibility of alternative translation initiation sites. To address this issue, we mutated the methionine at position 10 to alanine in the TgTPx1/2-HA construct and checked the localization of the protein. TgTPx1/2(M10A) co-localizes with the mitochondrial protein SPTP-SOD2-GFP and apicoplast protein ACP (Figs. 4B, 4C) and hence, is dually targeted to both the endosymbiotic organelles. This indicates that the dual targeting of TgTPx1/2 is not dependent on two different translation products from the methionines at positions 1 and 10. Therefore, a single translational product of TgTPx1/2 drives the dual targeting of the protein to the apicoplast and the mitochondrion.

## Bioinformatics analysis of the N-terminus of TgTPx1/2 predicts altered localization to the apicoplast and mitochondrion

A single translational product of TgTPx1/2 is dually targeted to the apicoplast and the mitochondrion. Additionally, the first 50 amino acids of the N-terminus of the protein appear to have ambiguous signals for this dual targeting. The first step of targeting to the apicoplast via the ER, is recognition of the signal peptide by the signal recognition particle, SRP. The first step of targeting for a majority of mitochondrial targeted proteins is the recognition of the MTS by translocons, therefore ambiguous signals can be attributed to the presence of both a signal peptide and MTS within the first 50 amino acid residues of TgTPx1/2. These targeting sequences are predicted by the algorithms SignalP 3.0-HMM and MitoProt. When TgTPx1/2 was subjected to analysis by these algorithms, the protein was seen to have a signal peptide probability score of 0.527 and a mitochondrial probability score of 0.98. Interestingly, the signal peptide score is much lower than the score of 0.994 and 0.731 observed for the classical apicoplast targeted acyl carrier protein (TgACP) and Ferredoxin-NADP$^+$ reductase (TgFNR) respectively.

Next, each amino acid residue of the first 50 amino acids of TgTPx1/2 was replaced *in silico* with an alanine. For each of these *in silico* alanine scanning mutants, we determined the SignalP 3.0-HMM and MitoProt scores. These scores were found to vary compared to the wild type protein between the stretches of 2–30 amino acids therefore data for this region is shown (Fig. 5). For mutants 2–14, there is no change in MitoProt scores, suggesting that these mutants should be localized to the mitochondrion. For the SignalP 3.0-HMM scores, variations were observed for many of the alanine mutants. At least one of these mutants has been assayed for its *in vivo* localization (M10A) and shows no alteration in localization compared to wild type TgTPx1/2 (Figs. 4B, 4C). Therefore, we suggest that a change in the SignalP score from 0.527 to 0.573 does not affect apicoplast localization. However, some mutants that lie in the region between 15–30 amino acids show changes in both the SignalP 3.0-HMM and MitoProt score. Although the MitoProt scores varied between 0.9 to 1.0, it was important to test whether these seemingly minor variations had an impact on the localization of mutant proteins *in vivo*. Three mutants were selected and the respective mutations generated in the TgTPx1/2 gene, followed by transfection and *in vivo* localization studies.

## A TgTPx1/2 mutant with a higher SignalP score and a lower MitoProt score is targeted to the apicoplast and not the mitochondrion

Positively charged amino acids such as arginine and lysine are known to play an important role for mitochondrial import of proteins (*Truscott, Brandner & Pfanner, 2003*; *Murcha et al., 2014*). In apicomplexans too, positively charged residues in the apicoplast and MTS have been reported to play an important role in apicoplast and mitochondrial trafficking of proteins (*Toursel et al., 2000*; *Brydges, 2003*; *Tonkin, Roos & McFadden, 2006*; *Tonkin et al., 2008*). Analysis of the N-terminal 50 amino acids of TgTPx1/2 reveals five positively charged amino acid residues but arginine at position 24 is the only positively charged amino acid in the signal peptide (amino acids 1-30) of TgTPx1/2. Interestingly, bioinformatics predictions showed that replacing arginine at position 24 with alanine (R24A) increased

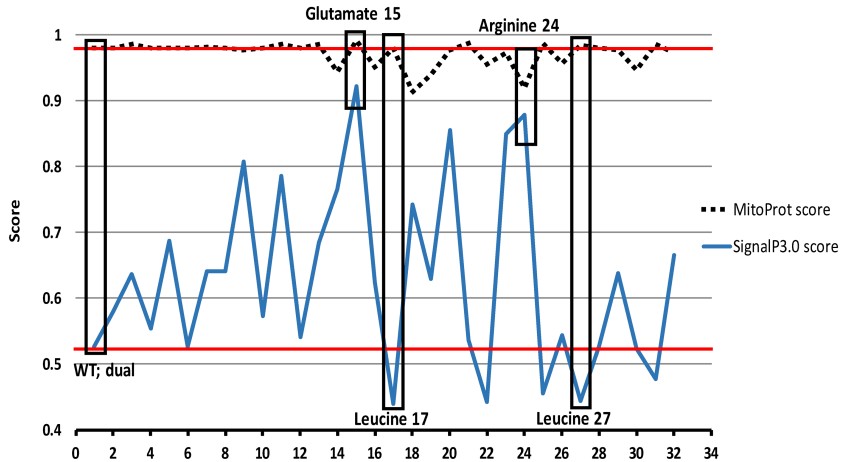

**Figure 5** **Bioinformatics analysis of the N-terminus of TgTPx1/2.** Determination of SignalP 3.0-HMM and MitoProt scores of alanine scanning mutation analysis of the N-terminus of TgTPx1/2. The full-length TgTPx1/2 protein was used as an input sequence in the two bioinformatics software. For each of the first 30 amino acids, the native amino acid was replaced with alanine and the resulting full length proteins were used as input sequence. Default settings were employed for the analysis. The SignalP 3.0-HMM (blue line) and MitoProt (dashed black line) scores were plotted for the WT and the mutant proteins. Red lines indicate the scores of SignaP3.0-HMM and MitoProt for the wildtype TgTPx1/2. Black boxes indicate the scores of SignalP 3.0-HMM and MitoProt of the mutants used in this study.

the signal peptide score from 0.527 to 0.878 and decreased the MitoProt score from 0.98 to 0.919 (Fig. 6A). Additionally for this mutant, an increase from 0.87 to 1.080 in the grand average of hydrophobicity (GRAVY) value (*Wilkins et al., 1999*) was seen on replacing a hydrophilic amino acid arginine with a hydrophobic amino acid alanine. The bioinformatics analysis collectively suggested that the TgTPx1/2(R24A) mutant should be localized exclusively to the apicoplast.

In parasites expressing TgTPx1/2(R24A), the mutant protein co-localized with ACP (Fig. 6B) with an average PCC of 0.73, indicating trafficking to the apicoplast. Furthermore, the mutant protein TgTPx1/2(R24A) did not co-localize with the mitochondrial marker MitoTracker Red (Fig. 6C) as evident by a negative average PCC of −0.05. This indicates that the change in the MitoProt score from 0.98 to 0.919 has a profound effect on mitochondrial localization of TgTPx1/2. Along with the apicoplast, some amount of the mutant protein localized to the ER. This was characterized by faint perinuclear staining (Fig. 6B), which has been observed in the case of other apicoplast targeted proteins as well (*Yung, Unnasch & Lang-Unnasch, 2001*; *Saito et al., 2008*).

Consistent with the bioinformatics predictions, increasing the SignalP score and decreasing the MitoProt score changed the dual localization of the protein TgTPx1/2 to a single organelle, the apicoplast. This suggests that a single point mutation that strengthens the signal sequence to a classical signal peptide favours apicoplast localization. Simultaneously, this mutation lowers the MitoProt score and abolishes mitochondrial trafficking completely, indicating the importance of the arginine residue for mitochondrial targeting of TgTPx1/2.

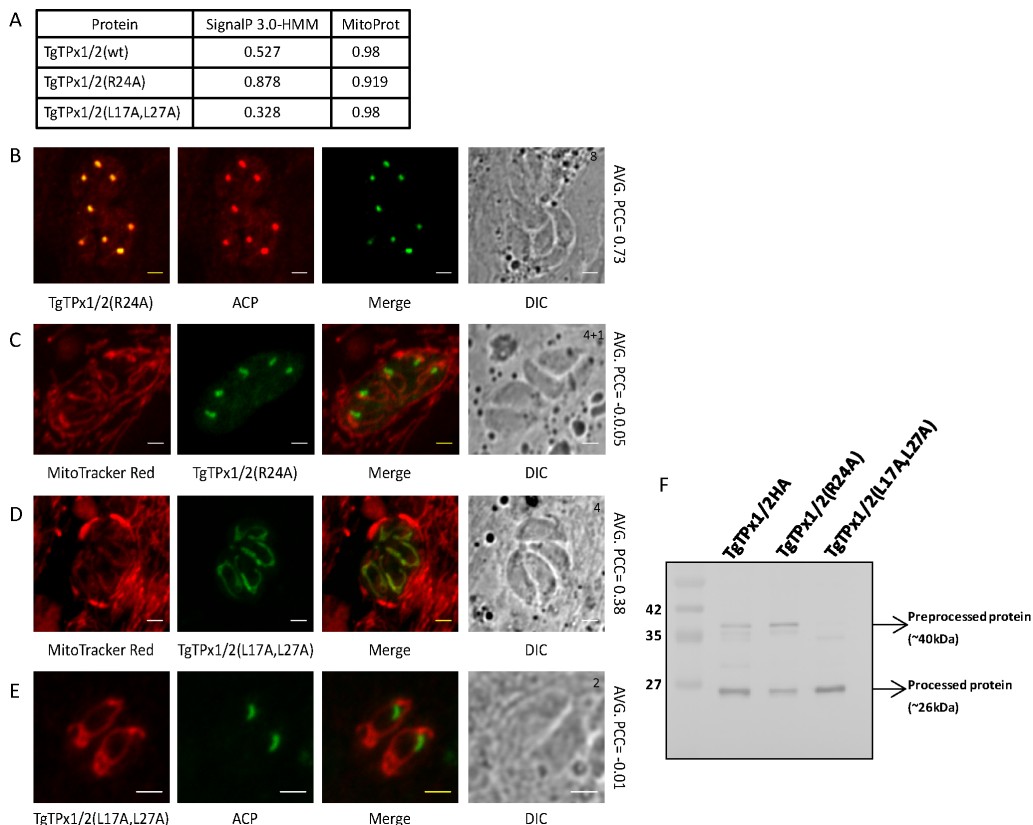

| Protein | SignalP 3.0-HMM | MitoProt |
|---|---|---|
| TgTPx1/2(wt) | 0.527 | 0.98 |
| TgTPx1/2(R24A) | 0.878 | 0.919 |
| TgTPx1/2(L17A,L27A) | 0.328 | 0.98 |

**Figure 6** **Analyzing the localization of TgTPx1/2 mutants, TgTPx1/2(R24A) and TgTPx1/2(L17A,L27A) in *T. gondii*.** (A) SignalP 3.0-HMM and MitoProt scores for the wildtype TgTPx1/2 and the mutants TgTPx1/2(R24A) and TgTPx1/2(L17A,L27A). Immunofluorescence images of the parasites expressing (B, C) TgTPx1/2(R24A) (red/green) and (D, E) TgTPx1/2(L17A,L27A) (green/red). Here ACP is used to label the apicoplast (green) while MitoTracker Red (red) is used for labelling the mitochondrion of *T. gondii*. Scale bar, two μm. Numbers (in black) at the top right corner of the DIC panel indicates the number of parasites inside the vacuole. (F) A Western blot analysis for the whole parasite lysates of wildtype TgTPx1/2, TgTPx1/2(R24A) and TgTPx1/2(L17A,L27A) using anti-HA antibodies. Full-length Western blot is represented in Fig. S2.

A Western blot was carried out of parasites stably expressing the TgTPx1/2(R24A) protein in the apicoplast. As observed for the wild-type protein TgTPx1/2, two bands of sizes ∼40 kDa and ∼26 kDa were detected (Fig. 6F). This indicates that the apicoplast-localized TgTPx1/2(R24A) contains two major protein species similar to those observed for the wild-type protein TgTPx1/2. Apart from the two intense bands, another faint band was observed for the R24 mutant protein and a similar faint band of the same size was also seen on the Western blot of the wild type protein TgTPx1/2, suggesting that this may be a pre-processed form of TgTPx1/2.

## A TgTPx1/2 mutant with a lower SignalP score and a high MitoProt score is localized to the mitochondrion and not to the apicoplast

Bioinformatics predictions of alanine scanning mutants showed two amino acids that increased the MitoProt score, while decreasing the SignalP score (L17A and L27A). Both

these leucine residues are present in the signal peptide region of TgTPx1/2. When these two leucines were simultaneously changed to alanines, the SignalP score decreased from 0.527 to 0.328 and the MitoProt score remain unchanged (Fig. 6A). Further bioinformatics analysis suggests that the GRAVY value of the signal peptide decreased from 0.87 to 0.73 on mutating the leucines at position 17 and 27 to alanines. These *in silico* predictions indicated that the TgTPx1/2(L17A,L27A) double mutant (Fig. 6A) should be present only in the mitochondrion.

Consistent with these bioinformatics analysis, in *T. gondii* parasites expressing TgTPx1/2(L17A,L27A), the mutant protein showed co-localization with a mitochondrial marker MitoTracker Red (Fig. 6D) with a PCC of 0.38 which is comparable to that of the wild type protein of 0.33. This mutant shows no overlapping staining with the apicoplast marker ACP (Fig. 6E) and the average PCC of −0.01 observed for these parasites is much lower than that observed for the wild type protein TgTPx1/2 (PCC 0.45). This clearly indicates that the protein localizes to the mitochondrion but not to the apicoplast.

To determine the size of the protein present in the mitochondrion, a Western blot of parasites stably expressing TgTPx1/2(L17A,L27A) was carried out using anti-HA antibodies. Unlike the wild-type TgTPx1/2, only a single intense processed band of ∼26 kDa was detected (Fig. 6F). This shows that the mitochondrial-localized TgTPx1/2 is found as a processed protein. Along with the intense band of 26 kDa, a faint band similar to the one seen for the wild type TgTPx1/2, was also observed. This may be a pre-processed version of the protein.

## A TgTPx1/2 mutant with higher SignalP and MitoProt scores is localized to the apicoplast

Charged residues in the targeting sequence of proteins have been reported to play an important role for mitochondrial and apicoplast trafficking of proteins (*Toursel et al., 2000*; *Brydges, 2003*; *Tonkin, Roos & McFadden, 2006*; *Tonkin et al., 2008*). Analysis of the first 30 amino acids revealed that along with a single positively charged amino acid arginine, a single negatively charged glutamate is present at position 15. Bioinformatics analysis indicated that changing the negatively charged amino acid glutamate residue increased the SignalP score of TgTPx1/2 from 0.527 to 0.922 while the MitoProt score changed from 0.98 to 0.99. On the other hand, simultaneous mutation of glutamate and arginine residues (E15A,R24A), also increased the SignalP score of TgTPx1/2 from 0.527 to 0.973 along with a change in the MitoProt score from 0.98 to 0.97 (Fig. 7A). As the signal peptide sequences would be highly strengthened with these mutations, while the MTSs would remain intact, one might expect to see dual localization of these mutant proteins. Additionally, mutating the only negatively charged amino acid glutamate in this stretch would confer a net positive charge on the signal peptide while mutating both the glutamate and arginine would restore the same net charge as observed for the wild type protein. All these factors collectively led us to test the localization of these TgTPx1/2 mutants.

Mutating the negatively charged residue glutamate at position 15 affected the trafficking to both the apicoplast and the mitochondrion, localizing the protein possibly to the cytosol and the ER (characterized by a perinuclear staining) in a majority of the population

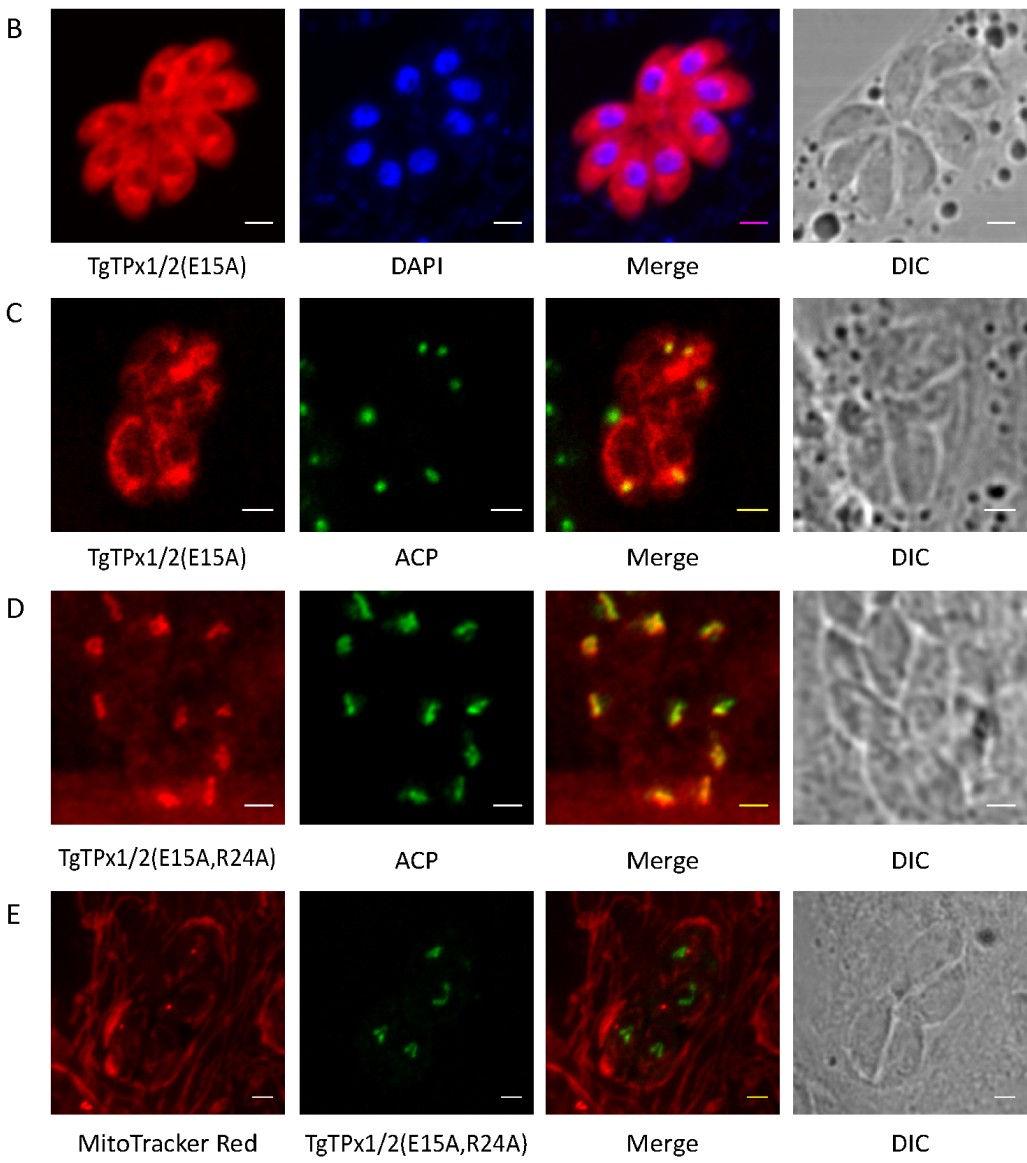

| Protein | SignalP 3.0-HMM | MitoProt |
|---|---|---|
| TgTPx1/2(wt) | 0.527 | 0.98 |
| TgTPx1/2(E15A) | 0.922 | 0.99 |
| TgTPx1/2(E15A,R24A) | 0.973 | 0.97 |

**Figure 7  Assessment of localization of TgTPx1/2(E15A) and TgTPx1/2(E15A,R24A), mutants of TgTPx1/2 with a higher SignalP 3.0-HMM and MitoProt scores.** (A) SignalP 3.0-HMM and MitoProt scores for the wildtype TgTPx1/2 and mutants TgTPx1/2(E15A), TgTPx1/2(E15A,R24A). Microscopic images of parasites transiently expressing (B, C) TgTPx1/2(E15A) (red) and (D, E) TgTPx1/2(E15A,R24A) (red/green). DAPI (blue) was used to label the parasite nucleus, MitoTracker Red (red) was used as a mitochondrial marker while ACP (green) was used to label the apicoplast. Scale bar, two μm.

(Fig. 7B). Only in a few parasite vacuoles, dual targeting characterized by a clear apicoplast and mitochondrial signal was observed (Fig. 7C).

Contrary to expectation, the double mutant (E15A,R24A), which displayed an increase in the SignalP and retained a high MitoProt score localized only to the apicoplast (Fig. 7D) with an average PCC of 0.65. This mutant restored the net charge on the signal peptide similar to that of the wild type protein. Along with the apicoplast, some amount of the mutant protein localized to the ER characterized by a faint perinuclear staining (Fig. 7D). No overlap with the mitochondrial marker MitoTracker Red (Fig. 7E; PCC of −0.02) further suggests that this mutant protein is not targeted to the mitochondrion. This result indicates that when a strong signal peptide is present with a strong MTS, the signal peptide dominates and the dual targeting of TgTPx1/2 is shifted entirely to the secretory route. Along with the strengths, we also observe that the charge on the signal peptide is a key determinant deciding the localization fate of the mutant proteins. Additionally, apart from the domination of signal peptide over the MTS, this phenotype can also be attributed to the mutation of arginine at position 24 to alanine which has been shown to play an important role in the mitochondrial targeting of the protein.

## TgTPx1/2 harbors an overlapping ambiguous stretch for dual targeting to the apicoplast and mitochondrion

Prediction and deletion analysis of the TgTPx1/2 N-terminus reveal that the first 30 amino acids encode a classical signal peptide while residues 30–50 harbor both a transit peptide and a MTS. The N-terminus of TgTPx1/2 was also analyzed for its propensity to form an α-helix. The α-helical wheel projection of the N-terminus predicts a hydrophobic helix from amino acid residues 1–30 (Fig. 8A). A hydrophobic helix is a characteristic feature of a signal peptide (*Kunze & Berger, 2015*) as observed for the signal peptide of TgACP (Fig. S3). Further, by carrying out *in silico* alanine scanning mutagenesis followed by point mutations we have delineated an overlapping region from 15–30 amino acids which encompasses essential amino acids responsible for dual targeting of TgTPx1/2 to the apicoplast and mitochondrion. Collectively based on the available data, α-helix prediction for the amino acid residues 15–50 was carried out. Interestingly, the 15–50 amino acid stretch forms a characteristic amphipathic helix (Fig. 8B). An amphipathic α-helix is a characteristic feature of a MTS (*Brydges, 2003*; *Zhu et al., 2004*; *Turk et al., 2013*) as observed for the mitochondrial-targeted heat shock protein 60 (TgHSP60) (Fig. S3). This indicates that the first 30 amino acids possess a characteristic hydrophobic helix for a signal peptide while the stretch from 15–50 amino acids has a propensity to form a characteristic amphipathic α-helix. Both these regions overlap (Fig. 8C), and we have shown by mutational analysis of amino acid residues (between 15–30) that this overlapping region plays an important role in the dual targeting of TgTPx1/2. Additionally, there might be residues in the stretch from 1–15 as well that might show an effect on the dual targeting of TgTPx1/2. Overall, we suggest that dual targeting of TgTPx1/2 can be explained by an ambiguous N-terminus having a signal peptide that overlaps with the MTS.

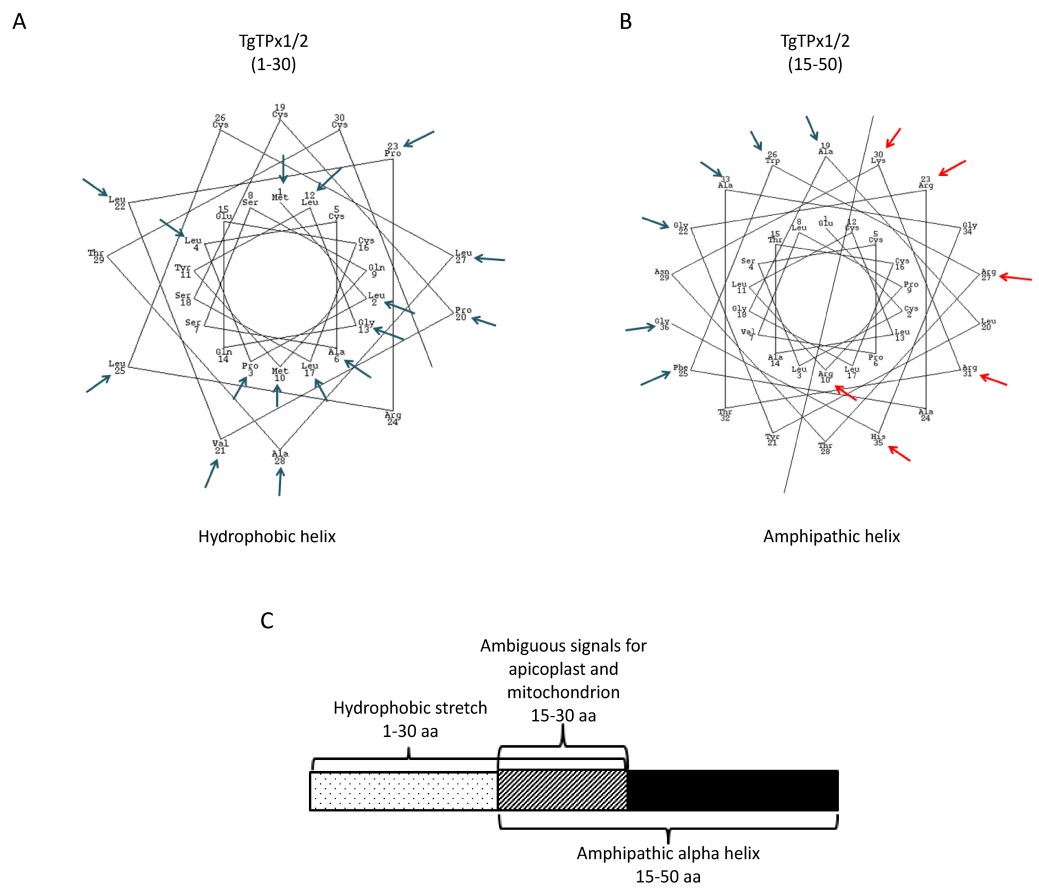

**Figure 8  Assessing the organellar targeting sequence for a dually targeted TgTPx1/2.** α-helix predictions for the (A) signal peptide and the (B) mitochondrial targeting sequence of TgTPx1/2 (https://www. webgenetics.com/acts/wg?group=protein&prog=wheel). The blue arrows mark the hydrophobic residues while the red arrows label the positively charged amino acids on the helical projections for the N-terminus of the proteins used for analysis. (C) A schematic representation of the N-terminal organellar targeting sequence of a dually targeted TgTPx1/2.

## DISCUSSION

### Targeting signals for organellar trafficking of TgTPx1/2 are overlapping yet separable

Import of proteins to endosymbiotic organelles requires specific targeting sequences that are present at their N-terminus (*Kunze & Berger, 2015*). These targeting sequences are specific for a particular organelle. A major aspect of this study was to delineate the targeting signals responsible for dual targeting of TgTPx1/2, an antioxidant protein, to the apicoplast and the mitochondrion in the apicomplexan parasite *T. gondii*.

In this study, by both bioinformatics and deletional analyses we have shown that the dually targeted protein TgTPx1/2 possesses a signal peptide at its N-terminus (region from residues 1-30). Previous studies have demonstrated that the signal peptides that bind to the signal recognition particle (SRP) for protein trafficking to the ER are enriched with leucine repeats (*Labaj et al., 2010*). The signal peptide of TgTPx1/2 has seven leucine residues and

as predicted by the bioinformatics analysis, mutating two of these leucine residues affected trafficking of TgTPx1/2 to the apicoplast, localizing the protein only to the mitochondrion. We suggest that mutating the leucine residues weakens or lowers the hydrophobicity of the signal peptide to an extent that it is not recognized by the SRP for targeting it to the ER so as to reach its final destination in the apicoplast.

The transit peptide directs the trafficking of the protein from the ER to the apicoplast, and this sequence is known to be enriched in positively charged amino acid residues (*Waller et al., 1998*; *Zuegge et al., 2001*; *Tonkin, Roos & McFadden, 2006*; *Tonkin et al., 2008*). Interestingly, MTSs also are enriched with the positively charged amino acid, arginine (*Von Heijne, Steppuhn & Herrmann, 1989*; *Lemire et al., 1989*; *Kunze & Berger, 2015*) which is believed to interact with the negatively charged region of the mitochondrial import receptor Tom22 for import of proteins (*Brix, Dietmeier & Pfanner, 1997*). As might be expected, Tom22, an integral component of the Tom translocon complex, has been identified and shown to be essential for mitochondrial import of nuclear-encoded proteins in *T. gondii* (*Van Dooren et al., 2016*). Therefore, mutation of the arginine residue at position 24 could have resulted in loss of either apicoplast or mitochondrial signal or both. The observation that the R24A mutant resulted in a loss of mitochondrial localization alone suggests that the transit peptide can tolerate amino acid changes more easily than the MTS of TgTPx1/2. Indeed, there are reports showing that transit peptides are dependent more on the amino acid charges present rather than specific sequences (*Tonkin, Roos & McFadden, 2006*; *Tonkin et al., 2008*).

Another interesting observation from this report is that the first 40 residues of TgTPx1/2 resulted in localization of the protein to punctate structures surrounding the apicoplast protein TgACP. These punctate structures might be similar to large vesicles that have been reported previously (*Bouchut et al., 2014*). The first 40 amino acids of TgTPx1/2 may interact with ER receptors that can carry the protein out of the ER, towards the apicoplast. In contrast, the first 50 residues of the protein were able to direct the protein to the apicoplast and mitochondrion. Apparently, the residues between 40–50 contain sequences that allow recognition by apicoplast membrane receptors to allow uptake into the organelle. Interestingly, upon examination of the sequence of amino acids in this region, **R**TN**KR**TAG**H**G, the presence of four positively charged residues is immediately apparent. These charged residues may be recognized by apicoplast receptors that interact with the protein in punctate structures that lie in close proximity to the organelle. This report adds to the existing literature (*Waller et al., 2000*; *DeRocher et al., 2000*; *Yung, Unnasch & Lang-Unnasch, 2001*; *Harb et al., 2004*; *Sheiner et al., 2015*) that shows a clear delineation of amino acid residues in the transit peptide for ER exit and apicoplast uptake of apicoplast targeted proteins.

Apart from apicoplast staining, the mitochondrial localization of TgTPx1/2N$_{1-50}$-EGFP indicates that the residues from 1–50 harbor a MTS as well. Bioinformatics and mutational analyses of this stretch suggest that the sequence from 15–50 is an internal MTS responsible for mitochondrial import of TgTPx1/2. There has been a previous report in *T. gondii* that suggests that an internal mitochondrial targeting signal assists in mitochondrial import of the protein (*Brydges, 2003*; *Garrison & Arrizabalaga, 2009*).

Further, Western blot analyses of the wildtype protein and mutants suggest that the processed proteins in apicoplast and mitochondrion are approximately the same size and may have cleavage sites that are identical or very close to each other. MitoProt analysis suggests that these cleavage sites are present at or near the 100th residue.

It is important to note that this study utilizes a non-native tubulin promoter to understand the phenomenon of dual targeting of TgTPx1/2, similar to that employed for studying TgSOD2 (*Pino et al., 2007*). Even though there are no reports of the tubulin promoter mis-targeting proteins to the apicoplast and mitochondrion, it would be interesting to study the dual targeting of TgTPx1/2 under its native promoter.

Thus, our study demonstrates that in TgTPx1/2, the signals for apicoplast (BTS) and mitochondrial (MTS) trafficking overlap. Previous reports have suggested that the transit peptides of apicoplast targeting proteins are similar to the MTS (*DeRocher et al., 2000*; *Yung, Unnasch & Lang-Unnasch, 2001*). We propose that these similarities in the organellar targeting sequences might have facilitated the superimposition of signals, rather than maintain separate distinct signals for dual targeting to these organelles in a cell. Additionally, mutation of amino acids in this overlapping stretch (residues 15–30) affects the dual targeting of the protein, localizing the protein either to the apicoplast or the mitochondrion. This indicates that despite the overlap of apicoplast and mitochondrial targeting signals, the residues for organellar targeting of TgTPx1/2 can be easily separated, directing the protein to a single organelle, thereby allowing the sampling of the two sub-cellular compartments with small changes in the N-terminus of the protein. These results have implications for the evolution of dually targeted proteins where it has been proposed that retaining the ability to sample multiple compartments confers an evolutionary advantage during the process of fixing the localization of nuclear–encoded organellar proteins (*Martin, 2010*).

### *In silico* and *in vitro* analyses suggest that the signal recognition particle (SRP) and the mitochondrial translocons compete for the ambiguous N-terminal signal sequence of TgTPx1/2

Proteins that localize to single organelles have highly specific signal sequences to avoid mis-localization; hence, organellar receptors have also evolved to recognize distinct signals with high stringency (*Kunze & Berger, 2015*). Against this backdrop, dual targeting of proteins to multiple organelles is particularly interesting. Apicoplast protein trafficking follows the secretory route where the nascent signal peptide of the protein is proposed to be recognized by the SRP and the protein is then trafficked to the apicoplast via the ER. On the other hand, mitochondrial trafficking takes place post-translationally and is mediated by translocons present on the mitochondrial membranes. Therefore, one might expect that dually targeted proteins would contain sequences that are recognized by both the SRP, for uptake into the ER, and by mitochondrial translocons. Data shown in this report for TgTPx1/2 are consistent with this hypothesis. It is important to note that, as recognition by the two receptors takes place in the cytosol, these receptors appear to compete for signals in the full length protein.

It has been proposed that there is a hierarchy of recognition by different receptors with the highest affinity being attributed to the SRP to direct signal peptide-containing proteins to the secretory route (*Kunze & Berger, 2015*). This is consistent with the results shown in this report. A mutant of TgTPx1/2 with a reduced signal peptide strength is localized exclusively to the mitochondrion and mutants with reduced MTS strength are localized exclusively to the apicoplast. However, a mutant of TgTPx1/2 with strong signal sequences for both organelles is localized preferentially to the apicoplast, consistent with the SRP having a higher affinity for the N-terminus ofTgTPx1/2 than the mitochondrial translocons. Given these results, it is tempting to speculate that the low score for the signal peptide of TgTPx1/2 is crucial for the dual localization of this protein.

This leads to a model that invokes competition between the SRP and mitochondrial translocons for the ambiguous N-terminal sequence of TgTPx1/2. As predicted by bioinformatics analysis, the strength of the signal peptide of TgTPx1/2 is lower, therefore, a subset of the nascent signal peptides might escape recognition by the SRP and continue to be synthesized on the cytosolic ribosomes. In the cytosol, the protein is recognized by mitochondrial translocons because of the presence of an N-terminal MTS. Therefore, we suggest that the ambiguity of the signal sequence combined with the hierarchy of protein trafficking drives the dual targeting of TgTPx1/2.

## CONCLUSION

A combination of bioinformatics and mutational analysis reveals that TgTPx1/2, a dually targeted protein of *T. gondii*, has an N-terminal sequence that consists of apicoplast and mitochondrial targeting signals superimposed upon one another. This ambiguous targeting sequence has an interesting property: the two targeting signals can be separated from each other by single amino acid mutations. We suggest that dually targeted proteins may possess N-terminal targeting sequences that allow them to sample multiple cellular compartments with small changes in the protein, a strategy that may have evolutionary benefits.

## ACKNOWLEDGEMENTS

We are grateful for the generous donations of ACP antibodies by Dhanasekaran Shanmugam, plasmids by Dominique Soldati-Favre and Shobhona Sharma and *T. gondii* strains by Shobhona Sharma. We thank Aparna Prasad for critical feedback and gratefully acknowledge Aishwarya Narayan for her expert comments on the manuscript. We thank the Industrial Research and Consultancy Centre (IRCC) at IIT Bombay for the confocal laser scanning microscopy facility.

### Funding

This work was supported by the Department of Biotechnology, India (No. BT/PR13546/BRB/10/1423/2015). Pragati Mastud received fellowship from the Government of India, Ministry of Human Resource Development (MHRD) and the

Indian Institute of Technology Bombay. The funders had no role in study design, data collection and analysis, decision to publish, or preparation of the manuscript.

### Grant Disclosures

The following grant information was disclosed by the authors:
Department of Biotechnology, India: No. BT/PR13546/BRB/10/1423/2015.
Government of India, Ministry of Human Resource Development (MHRD).
Indian Institute of Technology Bombay.

### Competing Interests

The authors declare there are no competing interests.

### Author Contributions

- Pragati Mastud conceived and designed the experiments, performed the experiments, analyzed the data, prepared figures and/or tables, authored or reviewed drafts of the paper, approved the final draft.
- Swati Patankar conceived and designed the experiments, analyzed the data, contributed reagents/materials/analysis tools, prepared figures and/or tables, authored or reviewed drafts of the paper, approved the final draft.

### Data Availability

   All data generated and analyzed during this study are included in the article and the Supplemental Files.

### Supplemental Information

Supplemental information for this article can be found online at http://dx.doi.org/10.7717/peerj.7215#supplemental-information.

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
