# Peer review of "An ambiguous N-terminus drives the dual targeting of an antioxidant protein Thioredoxin peroxidase (TgTPx1/2) to endosymbiotic organelles in Toxoplasma gondii"

_PeerJ, doi:10.7717/peerj.7215_

## Round 0.1 · original submission · Major Revisions

As you will see, both reviewers find the experimental work to be solid and valuable and I agree. There is some consensus that the choice of references could be better in a few cases, so I ask you to please have a look regarding citing the seminal reference or the most recent developments. Several specific suggestions are made. One reviewer took issue with the logical flow of the manuscript, I will leave the final choice to you but suggest that if it will stay as is you may want to clarify what is to be presented at the end of the introduction. I am fine with the bioinformatics remaining but it may be a good location to mention the update in prediction programs and their associated scores as mentioned by one reviewer. I do value the suggestion of reviewer 2 to address the criticisms of previous work. I am confident that while there are numerous suggested revisions, they are manageable.

·

Basic reporting

This manuscript by Mastud and Patankar is written in good English and generally makes use of the appropriate litterature references. That being said, I think the manuscript could be reorganised slightly and benefit from presenting the results in a more logical way (see my detailed comments to the authors).

Experimental design

The research question is well defined and most of the experimental approaches used seem valid and well controlled. However, a couple of strategies seemed a bit odd to me and would need to be clarified further (see my comments below).

Validity of the findings

Conclusions of the paper are generally well supported by the findings.

Additional comments

Main targeting signals for mitochondria, plastids and the ER share in their N-terminal position an α-helical structural element and subsequent removal from the core protein by intra-organellar cleavage. Dual targeting to endosymbiotic organelles through ambiguous targeting signals has been observed in plants previously (proteins localised to both mitochondria and chloroplasts) and is an interesting feature, especially from an evolutionary perspective. In Toxoplasma, there are some rare examples of dual protein localisations, like with antioxidant enzymes TgSOD2 and thioredoxin-dependent peroxidase TgTPX1/2 (Pino et al 2007). However, while the determinants of TgSOD2 targeting have been investigated by Pino et al, no precise determination of the sequence elements regulating organelle targeting has been performed for TgTPX1/2. The present manuscript aims at providing a detailed investigation of these sequence elements and is thus potentially interesting.

Main comments.

As a general comment on the organization of the manuscript, I feel that some the results are not presented in a logical way. Bioinformatic prediction of the signals should be used more clearly as a base for justifying subsequent experimental validation. For instance, for the part starting at l.234, it should be mentioned beforehand that a SignalIP analysis indicates a potential signal peptide with a putative cleavage site at AA 30. Mitoprot analysis should be also detailed, and not only the score but also the putative position of the potential cleavage site should be mentioned. Whether or not this fits with the observed molecular mass of the mature TgTPx1/2 as observed by western blot should be also discussed. Because as the manuscript currently stands, the rationale for choosing to express the first 50AA of the protein is unclear. If the Mitoprot analysis suggests this fragment could be encompassing the mitochondrial targeting signal, this should be more clearly mentioned.

I was very surprised when running the latest version of Mitoprot (Mitoprot II- v1.101) with TgTPx1/2, to see a much higher score (0.9762) and a putative cleavage site at AA100. Afterwards, I realized it is briefly mentioned in the ‘Materials & Methods’ section (l. 184-l.187). Any idea as to why there is such a discrepancy when using the latest algorithm of Mitoprot II? In any case, I thus do not think the authors can discuss extensively the relatively ‘low’ mitochondrial probability score by comparison with other known mitochondrial proteins (l. 519-521).

It is somewhat confusing that a dual Mitoprot/SignalIP analysis is conducted in parallel on the first 30AA (Fig. 4 and corresponding results). Although, this region can be both important for targeting to the apicoplast and to the mitochondrion, the sequence determinants for targeting to the mitochondrion likely involve a larger region, which is supported by experiments described in the current manuscript.

Related to this, the l. 67-76 part in the introduction should be written more clearly to distinguish between the typical apicoplast bipartite signal with ER type N-terminal secretory signal peptide followed by a plant-like transit peptide, and the mitochondrial targeting signal involving a single targeting signal, and which is generally independent from trafficking through the ER. In fact, many mitochondrial proteins even lack N-terminal pre-sequences and instead harbor internal signals to direct them to the mitochondrion. For the same reason, the authors should rephrase l. 300-301, ‘As the first steps of targeting for the apicoplast and mitochondrion are recognition by the ER and the mitochondrial translocons’, which is confusing.

A transit peptide has been potentially identified (l. 475), but it hasn’t been formerly experimentally verified to act as such. Thus, the authors shouldn’t write ‘This is the first report that shows a clear delineation of amino acids residues in the transit peptide…’ l. 477-478. Besides it seems to me some transit peptides have been identified previously, at least in Plasmodium. Again, the position of the transit peptide and the cleavage of the protein should be discussed further as the mature apicoplast and mitochondrial forms seem to have the same apparent molecular mass.

L. 398. I do not understand why the authors generated a double mutant for both E15 and R24. According to the results of the bioinformatics screen shown on Fig. 4E, a single E15A mutation is enough to increase both Mitoprot and SignalIP scores, while a single mutation R24A raises the SignalIP but decreases the Mitoprot score (and clearly affects mitochondrial targeting as seen on Fig. 5C). Although the authors claim prediction for the double mutant also show an increase in the Mitoscore score, I do not see the advantage of using this mutant over the simple E15A mutant. Given the apparent importance of the R24 residue for mitochondrial targeting, it might be wiser not to alter it in the context of this experiment. Instead, the authors should evaluate the localization of the single E15A mutation and see if it supports their hypothesis that increasing the two prediction scores favours apicoplast targeting over mitochondrial targeting.

Fig. 4 A-D could easily be merged to Fig. 7 and discussed altogether for the sake of simplification. Moreover, there is no proof the region containing the ambiguous signal is restricted to AA15-30. Fig. 4E identifies potential AA mutations affecting the Mitoprot score in region AA1-14.

Minor comments.

L. 206-207: ‘for tagging proteins in T. gondii, GFP fusions show altered organellar localization while small epitope tags do not’. This seems like a general statement, but it obviously depends on the protein. If the authors want to refer to TgTPx1/2 in particular, this should be mentioned precisely.

There is no denying the in silico alanine scanning strategy yielded interesting results and led to the identification and experimental validation of key residues. However, alanine has a short hydrophobic side chain (some signal peptides are actually quite alanine-rich), although it is clearly not the most hydrophobic AA. Thus, the rationale for using an in silico alanine scanning strategy to identify potential important residues for targeting in the first 30AA should thus be explained more carefully in the manuscript. As charge is also likely to be affected and the authors suspect part of the signal peptide is also involved in an aliphatic chain important for mitochondrial targeting, this should be mentioned.

Fig. 4B. Residues 19-20 do not seem to fit in the representation of the helix model.

The authors have to describe precisely in the ‘Materials & methods’ section how the Pearson’s correlation coefficient has been calculated. Some of the currently values are awkward. For example, for the signals displayed on Fig. 1B and Fig 2C, the PCC is unusually high (given that although 100% of the pixels of the apicoplast signal may be co-localising with the TgTPx1/2 signal, but the latter contains many additional pixels not colocalised with the apicoplast signal). In addition, when considering large field images with Mitotracker labelling (Fig. 3D,F, Fig. 5C,D, Fig. 6C) the very low values sometimes displayed suggest there could have processing of the whole field (including signal from the Mitotracker-labelled host mitochondria). A region of interest has to be defined for this type of analysis and background subtraction by thresholding may also be used.

An accession number for TgTPX1/2 should be mentioned in the “Materials & Methods” section.

Reviewer 2 ·

Basic reporting

Use of references should be extensively revised. Some specific examples for places where the used references are not suitable: the red algal origin of the apicoplast (line 63) and the references for SP/TP/MTS cleavage (line 68/69, 219). Likewise, missing key references e.g. Sheiner et al; Traffic 2015, lists most of the known targeting signals of apicoplast proteins in Toxoplasma (relevant for line 73/74), likewise, several reviews describe the metabolic roles of the two organelles (line 64).

Experimental design

Research question (basis of Tpx1/2 bimodal targeting) is well defined and experiments well designed and relevant to explore the features of the trafficking signal.

This is with one exception: I don’t see the value in the in silico result paragraph (296-333), I would suggest to remove it.

Validity of the findings

The findings are valid, however, conclusions should be drawn more carefully – examples: the mentioning of receptor competition (line 98-100) is an extrapolation; likewise claiming that 1-40-GFP fusion “escaped the ER” (line 258); stating 17% without providing statistically relevant approach and while not showing the data (line 283). There are more examples, proposed to revise conclusions throughout.

Additional comments

This is an interesting topic and one that is understudied and deserve more attention. In addition to the above comments, which are important to improve the clarity, depth and accuracy of the work, a more in-depth discussion in the intro about what we do and do not know about signals for dual targeted protein is needed (currently in lines 77-86).
For example, there are more described cases of dual targeting in Plasmodium (e.g. https://www.ncbi.nlm.nih.gov/pubmed/22889167 ) and in Toxoplasma (https://www.ncbi.nlm.nih.gov/pmc/articles/PMC3767399/). Likewise, this intro should discuss the different proposed mechanisms (alternative start sites, alternative splicing, overlapping amino-acid signals etc) and highlight work done to address those options (e.g. the Pino paper that is mentioned in other context and other work in Plasmodium). Finally, it is important to discuss criticism for previous work – e.g. SOD2 dual targeting was only seen under non-native promoter (and in fact the current study also expresses TPx1/2 under Tub promoter) – this should be highlighted.

---

## Round 0.2 · accepted · Accept

Thank you for your careful and detailed edits to the manuscript.

#